# HIV skews the SARS-CoV-2 B cell response towards an extrafollicular maturation pathway

Robert Krause[1,2]*, Jumari Snyman[1,2,3], Hwa Shi-Hsia[1,4], Daniel Muema[1,2,3], Farina Karim[1,2], Yashica Ganga[1], Abigail Ngoepe[1], Yenzekile Zungu[1,2], Inbal Gazy[2,5], Mallory Bernstein[1], Khadija Khan[1,2], Matilda Mazibuko[1], Ntombifuthi Mthabela[1], Dirhona Ramjit[1], COMMIT-KZN Team, Oliver Limbo[6], Joseph Jardine[6], Devin Sok[6], Ian A Wilson[7], Willem Hanekom[1,4], Alex Sigal[1,2,8,9], Henrik Kløverpris[1,4,10], Thumbi Ndung'u[1,2,3,4,8], Alasdair Leslie[1,4]

[1]Africa Health Research Institute, Durban, South Africa; [2]School of Laboratory Medicine and Medical Sciences, University of KwaZulu-Natal, Durban, South Africa; [3]HIV Pathogenesis Programme, The Doris Duke Medical Research Institute, University of KwaZulu Natal, Durban, South Africa; [4]Division of Infection and Immunity, University College London, London, United Kingdom; [5]KwaZulu-Natal Research Innovation and Sequencing Platform, Durban, South Africa; [6]International AIDS Vaccine Initiative, New York, United States; [7]The Scripps Research Institute, La Jolla, United States; [8]Max Planck Institute for Infection Biology, Berlin, Germany; [9]Centre for the AIDS Programme of Research in South Africa, Durban, South Africa; [10]Department of Immunology and Microbiology, University of Copenhagen, Copenhagen, Denmark

*For correspondence:
robert.krause@ahri.org

Group author details:
COMMIT-KZN Team See page 14

## Abstract

**Background:** HIV infection dysregulates the B cell compartment, affecting memory B cell formation and the antibody response to infection and vaccination. Understanding the B cell response to SARS-CoV-2 in people living with HIV (PLWH) may explain the increased morbidity, reduced vaccine efficacy, reduced clearance, and intra-host evolution of SARS-CoV-2 observed in some HIV-1 coinfections.

**Methods:** We compared B cell responses to COVID-19 in PLWH and HIV negative (HIV-ve) patients in a cohort recruited in Durban, South Africa, during the first pandemic wave in July 2020 using detailed flow cytometry phenotyping of longitudinal samples with markers of B cell maturation, homing, and regulatory features.

**Results:** This revealed a coordinated B cell response to COVID-19 that differed significantly between HIV-ve and PLWH. Memory B cells in PLWH displayed evidence of reduced germinal centre (GC) activity, homing capacity, and class-switching responses, with increased PD-L1 expression, and decreased Tfh frequency. This was mirrored by increased extrafollicular (EF) activity, with dynamic changes in activated double negative (DN2) and activated naïve B cells, which correlated with anti-RBD-titres in these individuals. An elevated SARS-CoV-2-specific EF response in PLWH was confirmed using viral spike and RBD bait proteins.

**Conclusions:** Despite similar disease severity, these trends were highest in participants with uncontrolled HIV, implicating HIV in driving these changes. EF B cell responses are rapid but give rise to lower affinity antibodies, less durable long-term memory, and reduced capacity to adapt to new variants. Further work is needed to determine the long-term effects of HIV on SARS-CoV-2 immunity, particularly as new variants emerge.

**Funding:** This work was supported by a grant from the Wellcome Trust to the Africa Health Research Institute (Wellcome Trust Strategic Core Award [grant number 201433/Z/16/Z]). Additional funding was received from the South African Department of Science and Innovation through the National Research Foundation (South African Research Chairs Initiative [grant number 64809]), and the Victor Daitz Foundation.

## Editor's evaluation

The authors investigate how HIV-1 infection affects the immune response in the context of SARS-CoV-2 infection by characterising the circulating B cell response. They conclude that people with HIV-1 infection, who become infected by SARS-CoV-2, produce B cell responses via an extra-follicular pathway to a greater degree than people who do not have HIV-1 infection. These findings imply that in HIV-1 infected individuals, long-term B cell and antibody responses against SARS-CoV-2 might not be as robust and durable compared to those in people without HIV-1 infection. The manuscript will be of interest to infectious disease specialists, virologists, and immunologists.

## Introduction

SARS-CoV-2 remains a threat to global health, especially in the light of new, more contagious variants capable of escaping vaccine-induced neutralizing antibodies (*Cele et al., 2021*; *Tada et al., 2021*; *Tegally et al., 2021*; *Wibmer et al., 2021*). Although vaccination may not prevent transmission, it is generally effective at preventing severe disease, primarily via eliciting neutralizing antibodies targeting the SARS-CoV-2 spike protein (*Frater et al., 2021*; *Shinde et al., 2021*; *Tada et al., 2021*). Risk factors for severe disease, especially in unvaccinated people, include old age (>65), underlying lung and heart disease; diabetes; and immune disorders such as those caused by HIV infection (*Williamson et al., 2020*). HIV has also been associated with increased morbidity and mortality (*Bhaskaran et al., 2021*; *Western Cape Department of Health in collaboration with the National Institute for Communicable Diseases, South Africa, 2021*), especially in patients with uncontrolled HIV viremia (*Chanda et al., 2021*) and in those with CD4 counts below 200 cells/µl, emphasizing the need for effective antiretroviral therapy (ART) (*Karim et al., 2021*). In addition, an inadequate immune response to COVID-19 is associated with prolonged SARS-CoV-2 infection and high intra-host mutation rates in both uncontrolled HIV and patients on immune-suppressing medication (*Cele et al., 2022*; *McCormick et al., 2021*). This highlights the importance of understanding the immune response to COVID-19 in patients with HIV, especially in the South African context, which has a high HIV prevalence (*Kharsany et al., 2018*) and SARS-CoV-2 attack rate (*Tegally et al., 2021*).

HIV affects the adaptive immune response by infecting CD4 T cells and reducing their numbers in circulation (*Dalgleish et al., 1984*; *Westendorp et al., 1995*). CD4 T follicular helper (Tfh) cells are a critical component of the germinal centre (GC) reaction as they assist the affinity maturation of their cognate B cell's antigen receptor (BCR). The knock-on effects of HIV infection can therefore include hypergammaglobulinemia (*Lane et al., 1983*), depleted resting memory, and increased naïve B cell frequencies (*Moir and Fauci, 2014*). Interestingly, as with other inflammatory diseases (*Freudenhammer et al., 2020*), HIV is also associated with an increased prevalence of 'tissue-like' memory B cells in circulation (*Ehrhardt et al., 2005*; *Knox et al., 2017*; *Moir and Fauci, 2014*). These CD27-ve CD21-ve B cells resemble the EF constituent now often referred to as double negative (DN) B cells (*Jenks et al., 2019*; *Jenks et al., 2020*; *Woodruff et al., 2020*). These HIV-induced changes in the B cell compartment are likely to be responsible for the reduced vaccine efficacy and durability observed in people living with HIV (PLWH), including to novel SARS-CoV-2 vaccines (*Hassold et al., 2022*; *Kernéis et al., 2014*), and may contribute to prolonged viremia and increased viral mutation (*Cele et al., 2022*; *Karim et al., 2021*).

We, therefore, investigated the effect of HIV on the B cell response to SARS-CoV-2 infection, using a comprehensive B cell phenotyping approach and B cell baits to identify SARS-CoV-2-specific B cells. Blood samples were collected at weekly intervals from PLWH and HIV-ve study participants following positive COVID-19 diagnosis during the first wave of infections in a cohort of patients from Durban, South Africa (*Karim et al., 2021*). In this cohort of individuals with predominantly mild-moderate

disease, the B cell response to SARS-CoV-2 infection is strikingly different in PLWH and characterized by reduced GC activity and a contrasting increase in EF activity.

## Methods

### Ethical approval

The study protocol was approved by the University of KwaZulu-Natal Biomedical Research Ethics Committee (approval BREC/00001275/2020). Written informed consent was obtained for all enrolled participants.

### Participant enrolment and clinical severity score

All study participants were over 18 years of age and capable of giving informed consent; presented with a positive SARS-CoV-2 diagnosis and were recruited from two hospitals (King Edward VIII or Clairwood) in Durban, KwaZulu-Natal, South Africa, between 8 June and 25 September 2020. In total 126 participants were enrolled. Participants consented to blood and nasopharyngeal/oropharyngeal swab collection at recruitment and during weekly follow-up visits. All participant SARS-CoV-2 diagnoses were verified by an in-house RT-qPCR test which also served to quantify the SARS-CoV-2 viral load. Two participants were excluded after their in-house RT-qPCR results remained negative and contradicted their initial diagnosis. All participants were ranked according to a clinical severity score of (1) asymptomatic, (2) symptomatic/mild without requiring supplemental oxygen, and (3) moderate requiring supplemental oxygen. A total of 10 healthy controls were included in the study that tested SARS-CoV-2 negative by PCR and were seronegative by ELISA (described below).

### Real-time qPCR detection of SARS-CoV-2

The QIAmp Viral RNA Mini kit (cat. 52906, QIAGEN, Hilden, Germany) was used according to the manufacturer's instructions to extract SARS-CoV-2 RNA from the combined nasopharyngeal and oropharyngeal swabs and 5 µl of the extracted RNA was used for RT-qPCRs. Three SARS-CoV-2 genes (ORF1ab, S, and N) were amplified using the TaqPath COVID-19 Combo kit and TaqPath COVID-19 CE-IVD RT-PCR kit (Thermo Fisher Scientific, Waltham, MA, USA) using a QuantStudio 7 Flex Real-Time PCR system and analysed using the Design and Analysis software (Thermo Fisher Scientific). Results were interpreted as positive if at least two of the three genes were amplified and regarded inconclusive if only one of the three genes was detected.

### Clinical laboratory testing

A separate blood sample per participant was sent to an accredited diagnostic laboratory (Molecular Diagnostic Services, Durban, South Africa) for HIV testing by rapid test and quantification of HIV viral load using the RealTime HIV-1 viral load test on an Abbott machine. A full blood count, including CD4 and CD8 count, was performed by another accredited diagnostic laboratory (Ampath, Durban, South Africa).

### Immune phenotyping of fresh PBMC by flow cytometry

Blood was collected in EDTA tubes and diluted 1 in 3 with PBS. Peripheral blood mononuclear cells (PBMC) were isolated by density gradient centrifugation through Histopaque 1077 (SIGMA) in SepMate separation tubes (STEMCELL Technologies, Vancouver Canada). For immune phenotyping $10^6$ fresh PBMC were surface stained in a 25 µl antibody mix containing a LIVE/DEAD fixable near-IR-dead cell staining reagent (1:200 dilution, cat. L10119, Invitrogen, Carlsbad, CA, USA) with combinations of the listed antibodies (*Supplementary file 1*) from BD Biosciences (Franklin Lakes, NJ, USA); or from BioLegend (San Diego, CA, USA) or from Beckman Coulter (Brea, CA, USA). Cells were stained for 20 min in the dark at 4°C, followed by two 1 ml washes with cold PBS, then fixed in 2% paraformaldehyde and stored at 4°C until acquisition on a FACSAria Fusion III flow cytometer (BD). Flow cytometry data was analysed with FlowJo version 9.9.6 (Tree Star).

### IgM and IgG ELISA detecting RBD-specific antibodies

Patient plasma samples were tested for the presence of anti-SARS-CoV-2 reactive IgM or IgG antibodies as described previously (*Snyman et al., 2021*). ELISA plates were coated with 500 ng/ml of

the D614G ancestral virus receptor binding domain (RBD) (GenBank: MN975262; provided by Dr Galit Alter, Ragon Institute, Cambridge, MA, USA) overnight at 4°C. Then blocked with 1% BSA-TBS at room temperature (RT) for 1 hr, followed by samples diluted at 1:100 in BSA-TBS+0.05% Tween 20 for 1 hr at RT. Secondary anti-IgM or -IgG antibodies (Jackson ImmunoReasearch, West Grove, PA, USA) were added at 1:5000 diluted in BSA-TBS+0.05% Tween 20 and incubated again for 1 hr at RT. Finally, plates were developed with one-step Ultra TMB substrate (Thermo Fisher Scientific) for 3 or 5 min respectively and signal development was stopped with the addition of 1 N $H_2SO_4$. Plates were washed with TBS+0.05% Tween 20 between each incubation step. All signals were compared to anti-SARS-CoV-2-specific monoclonal IgG (clone CR3022) or IgM (clone hIgM2001). Pre-pandemic plasma samples were used as negative controls to determine seroconversion cut-offs calculated as three times the standard deviation plus the mean.

## Statistical analysis

All analyses were performed in Prism (version 9; GraphPad Software Inc, San Diego, CA, USA). Non-parametric tests were used throughout, with Mann-Whitney and Wilcoxon tests used for unmatched and paired samples, respectively. Kruskal-Wallis H test was used for multiple comparisons. p Values less than 0.05 were considered statistically significant and denoted by *≤0.05; **<0.01; ***<0.001, and ****<0.0001.

## Results

We investigated the longitudinal dynamics of the B cell response to SARS-CoV-2 infection in PLWH compared to HIV-ve patients using a previously described COVID-19 cohort enrolled in Durban, South Africa, during the first wave of the pandemic in July 2020 (*Karim et al., 2021*). A total of 70 SARS-CoV-2 positive, confirmed by qPCR and serology, and 10 negative control participants were included in this study. Of the SARS-CoV-2-positive participants, 28 (40%) were PLWH, and 5 (18%) had detectable HIV in their plasma. SARS-CoV-2 infected participants were monitored weekly for five follow-up time points. Control participants were recruited at a single time point and were confirmed as SARS-CoV-2 negative by qPCR and serology and included two PLWH individuals. As some of the participants remained asymptomatic throughout the study, a timescale relating to days after positive diagnostic swab rather than symptom onset was used (days post-diagnostic swab), which has been shown to correlate well with symptom onset in the symptomatic patients from this cohort (*Karim et al., 2021*).

B cells were initially identified as CD19+ lymphocytes (*Figure 1A*), and the expression pattern of CD27 and CD38 was used to identify the canonical naïve, memory, and antibody-secreting cell (ASC) subsets (*Glass et al., 2020*). The ASC population was further differentiated into CD138+ plasma cells (PC) and CD138- plasmablasts (PB) (*Glass et al., 2020*). Considering all time points, HIV viremic participants had lower absolute B cell numbers, although this difference did not reach statistical significance. However, HIV-ve SARS-CoV-2 infected participants had significantly fewer naïve B cells and more memory B cells than SARS-CoV-2 infected PLWH and SARS-CoV-2 uninfected controls (*Figure 1B*; *Moir and Fauci, 2014*). The ASC response was significantly elevated at SARS-CoV-2 viremic time points in both patient groups, consistent with a robust ASC response to active infection (*Figure 1C*) but was not significantly different in PLWH. In addition, although the frequency of ASC at the earliest time point tended to be higher in the HIV-ve group, this was not statistically significant. Using the patient neutrophil-lymphocyte ratio (NLR) as a proxy of inflammation (*Ciccullo et al., 2020*; *Fu et al., 2020*; *Karim et al., 2021*), the frequency of ASC was found to be significantly higher at time points when the NLR ratio was above 3 (*Figure 1D*), suggesting an association between disease severity and ASC frequency. Finally, the PC:PB ratio was not significantly different between PLWH and HIV-ve patients at any time point, although there was a trend for a lower ratio in PLWH (*Figure 1E*). Taken together these data suggest that despite differences in canonical B cell phenotypes known to precede ASC maturation, PLWH mounted a similar ASC response against SARS-CoV-2 infection to HIV-ve participants.

To further investigate the B cell response to SARS-CoV-2 in PLWH, we designed three B cell phenotyping flow cytometry panels, using markers specific for B cell maturation, activation, homing, and regulatory function (*Figure 2A, C and D*). Fresh PBMC from all participants were stained using all three panels separately and analysed using an unbiased approach combining FlowSOM and tSNE

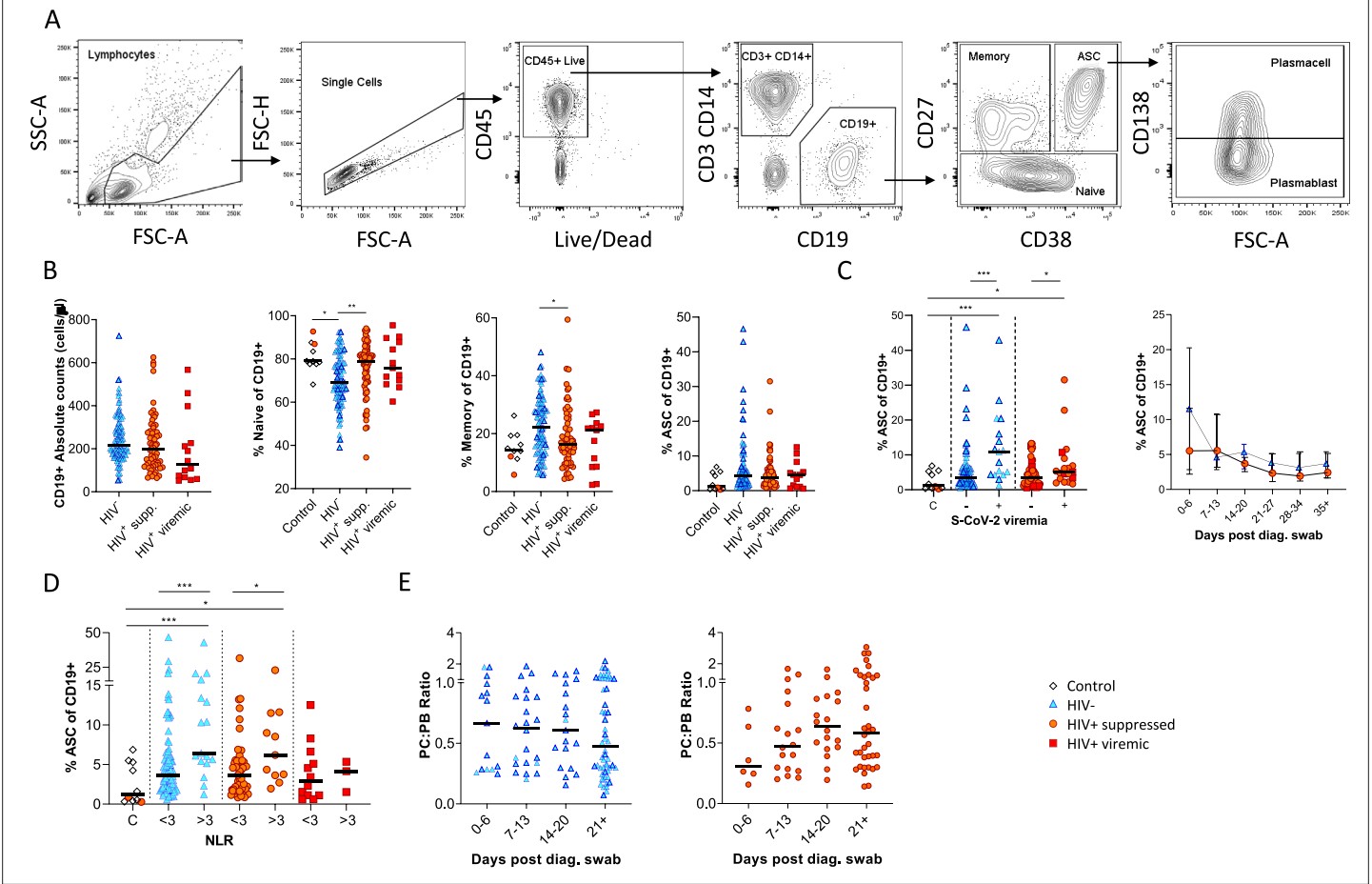

**Figure 1.** Canonical B cell phenotype frequencies vary with HIV status but still mount robust antibody-secreting cell (ASC) responses. (**A**) Gating strategy to identify CD19+ B cells within the peripheral blood mononuclear cell (PBMC) compartment. B cells were further gated on CD27 and CD38 to identify CD27+ Memory, CD27- Naïve, and CD27+CD38++ ASC. (**B**) Absolute B cell counts were calculated from patient total lymphocyte counts, followed by percent Naïve, Memory, and ASC fractions of the CD19+ parent population. These data represent the total combined frequencies over all time points per patient group. (**C**) The ASC response associated with SARS-CoV-2 viremia and was tracked longitudinally up to day 35 post diagnosis. Error bars indicate standard deviation (HIV- n = 37; HIV+ n = 33). (**D**) A neutrophil-lymphocyte ratio (NLR) served as a proxy of inflammation and associated with the ASC as well as plasmablast and plasma cell responses. Statistical analyses were performed using the Kruskal-Wallis H test for multiple comparisons and Mann-Whitney for SARS-CoV-2 viremia or NLR comparisons within groups. p Values are denoted by *≤0.05; **<0.01; ***<0.001, and ****<0.0001.

The online version of this article includes the following figure supplement(s) for figure 1:

**Figure supplement 1.** Individual patient longitudinal antibody-secreting cell (ASC) responses.

pipelines (*van der Maaten, 2008*; *Van Gassen et al., 2015*). This identified 11–12 distinct B cell clusters between the three phenotyping panels, with probable B cell phenotypes assigned based on the expression pattern of surface markers associated with each B cell cluster (*Figure 2A, C and D*). To uncover associations between HIV infection status or disease severity, an equal number of B cells for each disease severity category (ordinal scale 1–3 [OS1 to 3], *Karim et al., 2021*) were included, distributed equally between PLWH and HIV-ve controls. All comparisons in this figure were done with baseline samples. This allowed the relative abundance of each B cell cluster to be compared between the following clinical parameters: an ordinal disease scale (1–3), NLR as a measure of inflammation (*Ciccullo et al., 2020*; *Fu et al., 2020*; *Karim et al., 2021*), SARS-CoV-2 viral load, and HIV status.

Using B cell panel 1 (*Supplementary file 1*, *Figure 2A*), distinct patterns of homing marker expression were observed. Two CD27+CD38++ ASC populations are identified, highlighted in red and purple (*Figure 2A*). The former also expressed high levels of CXCR3, the primary receptor for CXCL9, -10, -11, which facilitates homing to inflamed tissues (*Onodera et al., 2012*; *Serre et al., 2012*; *Sutton et al., 2021*), high levels of the extravasation marker CD62L+, and, uniquely, the activation/

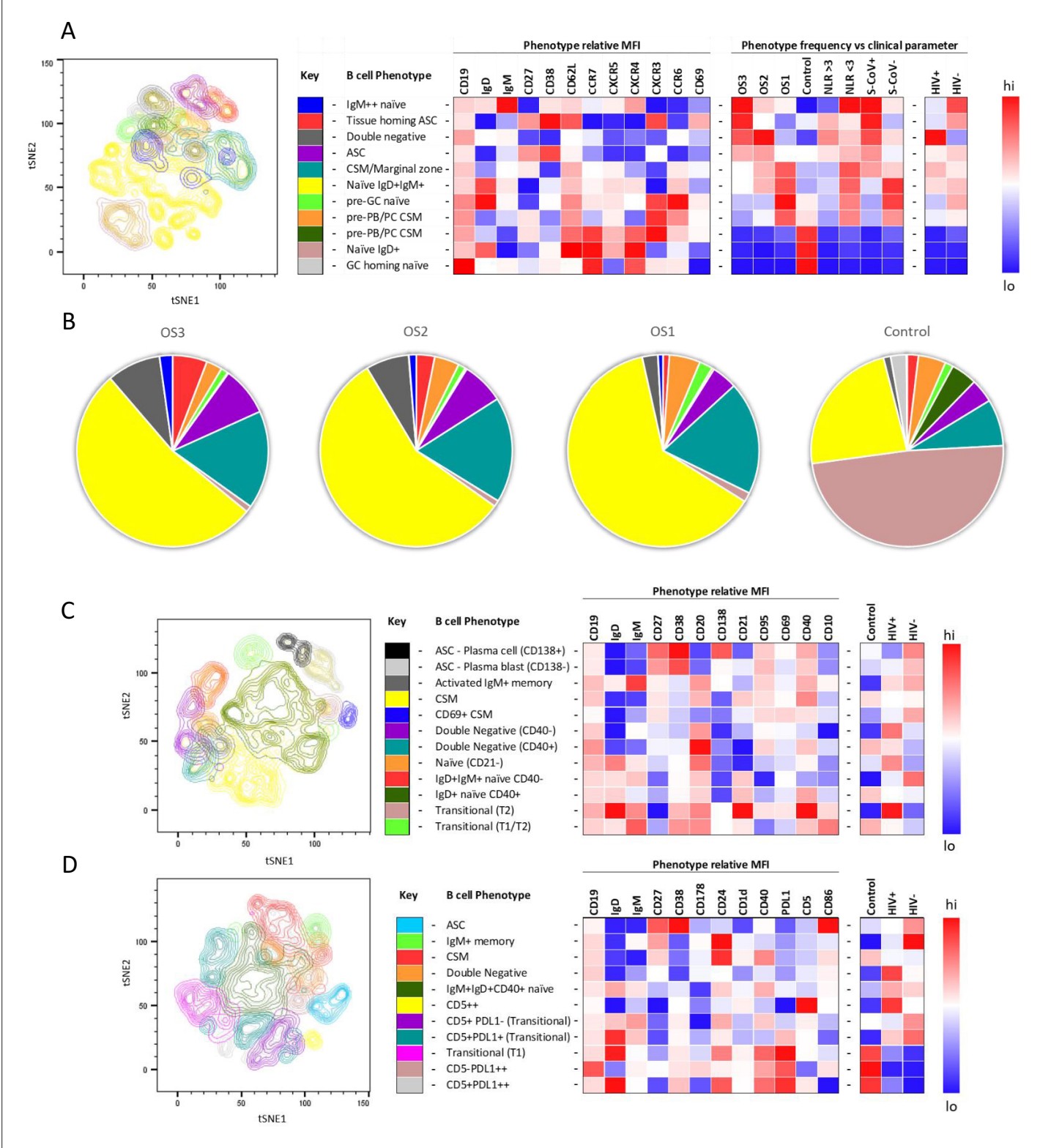

**Figure 2.** tSNE analysis of the B cell phenotypes and frequencies relative to COVID-19 clinical parameters including disease severity, neutrophil lymphocyte ratio, SARS-CoV-2 viremia, and HIV status. A total of 80,000 CD19+ B cells, all sampled at baseline, from four patient groups (20,000 cells per group) were used in an unbiased tSNE analysis. Patients were grouped by decreasing disease severity according to an ordinal scale ranging from 3 to 1 (OS3 to 1) and a healthy control group. There was an equal contribution from people living with HIV (PLWH) and HIV-ve patients per group except for the control group. Three B cell phenotyping panels were used, focusing on homing (**A, B**), maturation (**C**), and regulatory (**D**) markers. Each panel

*Figure 2 continued on next page*

*Figure 2 continued*

included an anchor panel of CD19, IgD, IgM, CD27, and CD38. The key to the colouring of the different tSNE clusters is included alongside with a description of the phenotype. The phenotype is then depicted as a heatmap of the median fluorescence intensity (MFI) of each surface marker within that cluster, followed by a heatmap of the cluster frequency relative to ordinal scale (OS3 to 1); neutrophil-lymphocyte ratio (NLR) cut-off of 3.0 to separate moderate and mild inflammation, SARS-CoV-2 viremia (S-CoV), and HIV status. (**B**) The frequency of each B cell phenotypic cluster identified by tSNE depicted as pie charts, separated based on disease severity (OS3 to 1) and controls.

The online version of this article includes the following figure supplement(s) for figure 2:

**Figure supplement 1.** Heatmap overlays B cell compartment tSNE plots focusing on homing markers.

**Figure supplement 2.** Heatmap overlays B cell compartment tSNE plots focusing on maturation markers.

**Figure supplement 3.** Heatmap overlays B cell compartment tSNE plots focusing on regulatory markers.

tissue residency marker CD69+. Therefore, this population is referred to as tissue homing and potentially indicates B cells which preferentially migrate to the diseased lung (*Onodera et al., 2012*; *Serre et al., 2012*; *Sutton et al., 2021*; *Weisel et al., 2020*). In line with this, tissue homing ASC are highly elevated in participants with the most severe disease (OS3), with elevated NLR, and with detectable SARS-CoV-2 viremia, as, to a lesser extent, are ASCs lacking this tissue homing phenotype (*Figure 2A and B*). In addition, a population of IgM^hi naïve B cells (highlighted in blue) was associated with severe disease, which co-expressed markers associated with GC homing (CXCR5 and CXCR4; *Cyster and Allen, 2019*). Strikingly, all three populations are elevated in HIV-ve participants but not in PLWH, as indicated in the final two columns of the heatmap. In PLWH, by contrast, disease severity was associated with an elevated population of B cells expressing a CD27-IgD- phenotype, corresponding to DN B cells (highlighted in grey), which lacked elevated expression of any of the homing markers measured. Several B cell subsets are elevated in asymptomatic subjects, including transitional B cells, as reported elsewhere (*Woodruff et al., 2020*) and a class switched memory (CSM) phenotype that expressed CXCR3, suggestive of differentiation into plasma blasts/cells (*Muehlinghaus et al., 2005*). As expected, in the control patients, the naïve B cell phenotype was predominantly IgD+ IgM- (brown), whereas the SARS-CoV-2 infected participants IgD+ naïve cells also expressed low levels of IgM+ (yellow; *Figure 2B*). Both naïve populations are high for CXCR5 and CXCR4, consistent with their requirement to gain entry to GC for affinity maturation (dark and light zones, respectively [*Cyster and Allen, 2019*]). Interestingly CCR6 was also upregulated in the SARS-CoV-2+ naïve B cells, suggestive of systemic B cell activation and transition of activated naïve B cells to pre-GC B cells (*Reimer et al., 2017*; *Schwickert et al., 2011*; *Wiede et al., 2013*). Together these data confirm that, as reported elsewhere (*Woodruff et al., 2020*), COVID-19 severity is associated with skewing of B cell phenotype and, for the first time, that this skewing is altered by concurrent HIV infection.

The association between disease severity and B cell phenotype was less apparent in data generated using the other two flow cytometry panels and, therefore not reported in the heatmaps (*Figure 2C and D*; *Figure 2—figure supplements 2 and 3*). However, distinct differences between PLHW and HIV-ve participants were also apparent. Panel 2, composed of maturation markers (*Figure 2C*), again suggests that ASC were less prevalent in PLWH than in HIV-ve participants, particularly CD138+ PC. In addition, CSM cells expressing CD69, consistent with activation, were elevated in HIV-ve participants but not PLWH. In contrast, two DN populations can be distinguished, differing by CD40 expression (purple and turquoise), both of which feed into an EF B cell maturation pathway (*Jenks et al., 2019*; *Jenks et al., 2020*; *Woodruff et al., 2020*), and both were elevated in PLWH. Likewise, an activated naïve (CD21^lo) population (orange) was also elevated in PLWH relative to HIV-ve participants. Two other naïve phenotypes were apparent, differing by their expression of CD40, of which the CD40-ve population appeared to be unique to the HIV-ve COVID-19+ participants (red population). Finally, two small transitional B cell populations were detectable, one of which, with a distinctive high IgD, CD21, and CD40 phenotype, was elevated in PLWH.

For panel 3, comprised of regulatory markers, the same reduced ASC response in PLWH was suggested (highlighted in blue), as the key B cell markers CD19, IgD, IgM, CD27, and CD38 are shared between all three panels. In this case, ASC are shown to express high levels of CD86, associated with B cell activation (*Cyster and Allen, 2019*). The same association between PLWH and DN B cells was also apparent (highlighted in orange), and which also display CD86 expression (*Figure 2D*). In addition, a small population of B cells expressing high levels of CD5 (CD5++) was elevated in PLWH. These

cells co-express CD1d, consistent with regulatory B cells producing the critical regulatory cytokine IL-10 (*Oleinika et al., 2018*; *Palmer et al., 2015*; *Yanaba et al., 2008*). Another important regulatory molecule in B cells is PD-L1, which can limit T-cell help via engaging with PD-1 on the surface of Tfh cells in the GC (*Khan et al., 2015*). The expression of this marker was primarily associated with B cell subsets present in non-COVID-19 controls (*Figure 2D*, pink and grey), suggesting it is downregulated during SARS-CoV-2 infection. However, PD-L1 expression did not appear to be affected by HIV coinfection. Taken together, these unbiased analyses consistently show that the B cell response to SARS-CoV-2 is skewed in PLWH, associated with a reduction in ASC subsets, class switching, and markers of GC homing and with an increase of B cell phenotypes associated with EF maturation.

To drill down further, we next analysed the skewed B cell subsets of interest in longitudinal samples by Boolean gating. Unlike in the tSNE analysis, where downsampling was used to prevent bias analysis, all study subjects were included in these analyses. Consistent with the potential difference in GC activity and class switching observed above, we found that CD27+ IgD- switched memory (SM) B cells were significantly more frequent in HIV-ve COVID-19 patients than in PLWH (*Figure 3A*). Furthermore, the fraction of SM B cells expressing CD62L and CXCR5, allowing them to home the GC, was reduced in PLWH, particularly in individuals with viremic HIV (*Figure 3B*). Longitudinal analysis shows this population increases over time in both groups, consistent with a dynamic change associated with SARS-CoV-2 infection but is consistently lower in PLWH (*Figure 3B*). A similar trend was observed for CSM B cells, which were expanded in HIV-ve participants compared to controls but were significantly lower in viremic PLWH (*Figure 3C*). Interestingly, IgM-only memory B cells (*Bautista et al., 2020*) were significantly upregulated in all COVID-19 patient groups relative to the control group, but again this was reduced in the PLWH group. Although the precise function of this B cell subset is debated, it is believed they can achieve rapid PC differentiation, GC re-initiation, and IgM and IgG memory pool replenishment (*Weill and Reynaud, 2020*). Together these data suggest that concurrent HIV infection may cause a reduction in GC homing, class switching, and memory establishment after SARS-CoV-2 infection, which was generally exacerbated in individuals with viremic HIV.

Next, given that the alternative to B cell maturation in the GC involves an EF route, DN B cells were examined in detail. The DN phenotype, also referred to as atypical B cells, forms part of an EF B cell response, which circumvents/pre-empts the GC reaction resulting in a rapid but short-lived PB response that facilitates rapid antibody production (*Jenks et al., 2019*; *Jenks et al., 2020*; *Song et al., 2022*; *Woodruff et al., 2020*). Both activated DN B cells, commonly referred to as the DN2 cells, and activated naïve phenotypes, contribute to the EF response (*Jenks et al., 2019*) and are identified as CD21$^{lo}$ CXCR5$^{lo}$ CD95+ subsets (*Figure 4A*; *Figure 4—figure supplement 1*). Multiple studies have described the expansion of DN2 B cells in association with severe COVID disease (*Chen et al., 2020*; *Kaneko et al., 2020*; *Woodruff et al., 2020*), but not in relation to HIV. Here, DN2 and activated naïve B cells were identified by expression of CD95, a known marker of activation on B cells (*Freudenhammer et al., 2020*; *Glass et al., 2020*; *Jenks et al., 2020*; *Le Gallo et al., 2017*). The DN2 phenotype studied here lacked both CD21 and CXCR5 expression ($R^2$=0.89, p<0.0001; *Figure 4—figure supplement 1A*), with a positive correlation for CD95 and CD11c expression ($R^2$=0.66, p<0.0001; *Figure 4—figure supplement 1B*). Both EF-associated B cell phenotypes were significantly more frequent in the SARS-CoV-2 infected PLWH relative to the HIV-ve group, irrespective of HIV viremia in the case of DN2 (*Figure 4B*; *Figure 4—figure supplement 2A* (i)). In addition, both subsets change in frequency over the course of infection, expanding from time point 2 (days 7–13) and remaining significantly expanded compared to HIV-ve participants until after time point 4 (days 21–27), after which they contract to the same level (*Figure 4C*; *Figure 4—figure supplement 2B*). The dynamic nature of this EF response strongly suggests that it emerges during SARS-CoV-2 infection and does not represent a pre-existing difference associated with HIV. Interestingly though, a direct association with SARS-CoV-2 viral load was evident only in the PLWH with suppressed HIV, whereas for HIV viremic individuals, the DN2 phenotype associated better with disease severity and inflammation (*Figure 4—figure supplement 1C*), which has been associated with chronic HIV infection (*Austin et al., 2019*). This is further supported by the fact that the frequency of DN2 and, to a lesser extent, activated naïve B cells, correlates with the RBD antibody titre in PLWH, particularly those with detectable HIV (*Figure 4D*; *Figure 4—figure supplement 2C*). In addition, the frequency of these populations was associated with increased clinical disease severity and NLR in PLWH but not in HIV-ve participants (*Figure 4C*; *Figure 4—figure supplement 2B*). Indeed, the lack of an EF response

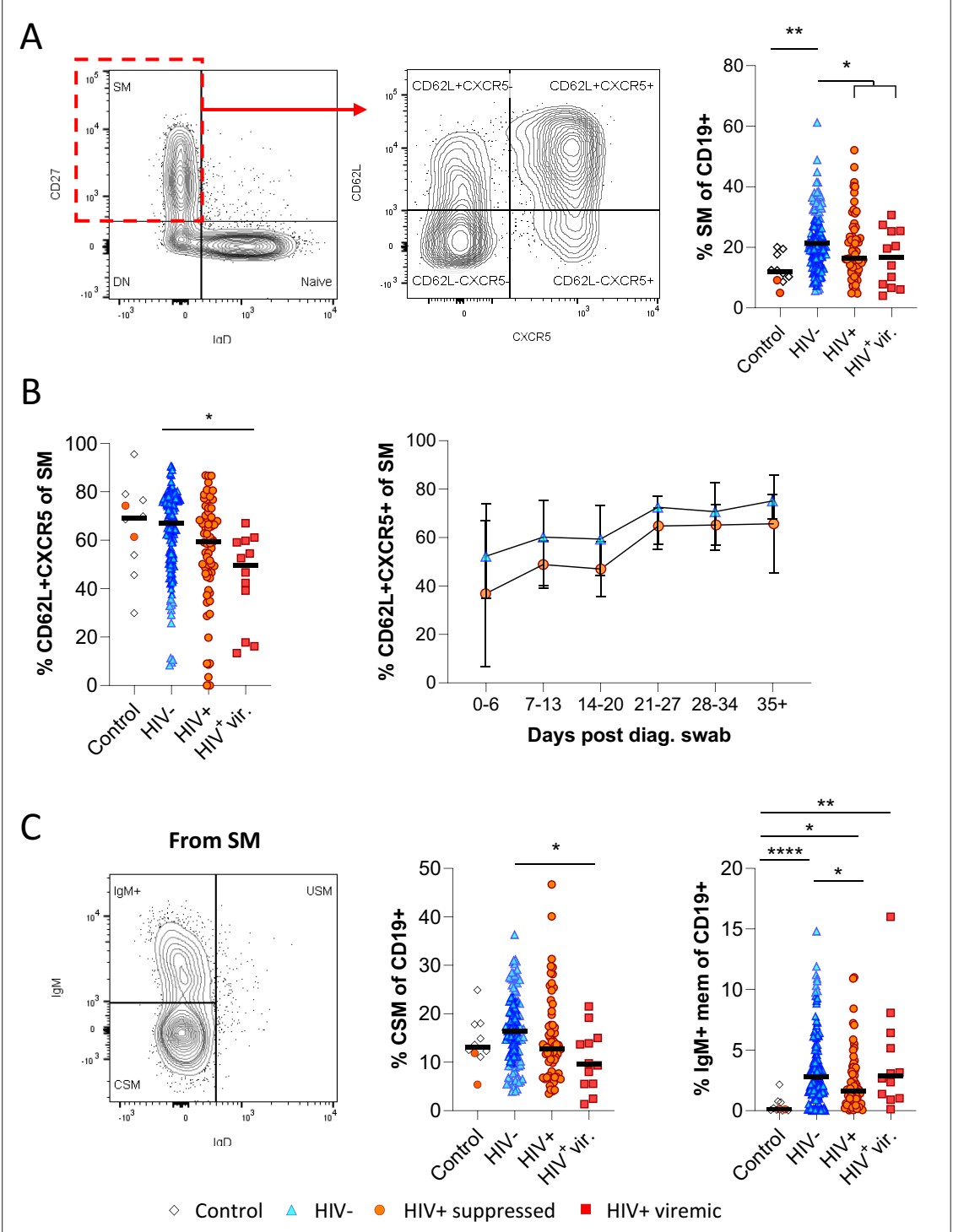

**Figure 3.** Reduced germinal centre homing and class switching of memory B cells in HIV viremic COVID-19 patients. (**A**) Gating strategy for total switched memory (SM; CD27+ IgD-) and homing to germinal centres (CD62L+ CXCR5+) and comparison of SM with respect to HIV status. (**B**) Germinal centre homing capacity relative to HIV status and longitudinal comparison. Error bars indicate standard deviation (HIV- n = 37; HIV+ n = 33). (**C**) The SM was further gated on IgM and IgD to identify IgM+ memory and class switched (IgM-IgD-) B cells. Both responses were compared with respect to HIV status. Statistical analyses were performed using the Kruskal-Wallis H test for multiple comparisons. p Values are denoted by *≤0.05; **<0.01; ***<0.001, and ****<0.0001.

The online version of this article includes the following figure supplement(s) for figure 3:

**Figure supplement 1.** Individual patient longitudinal switched memory germinal centre (GC) homing responses.

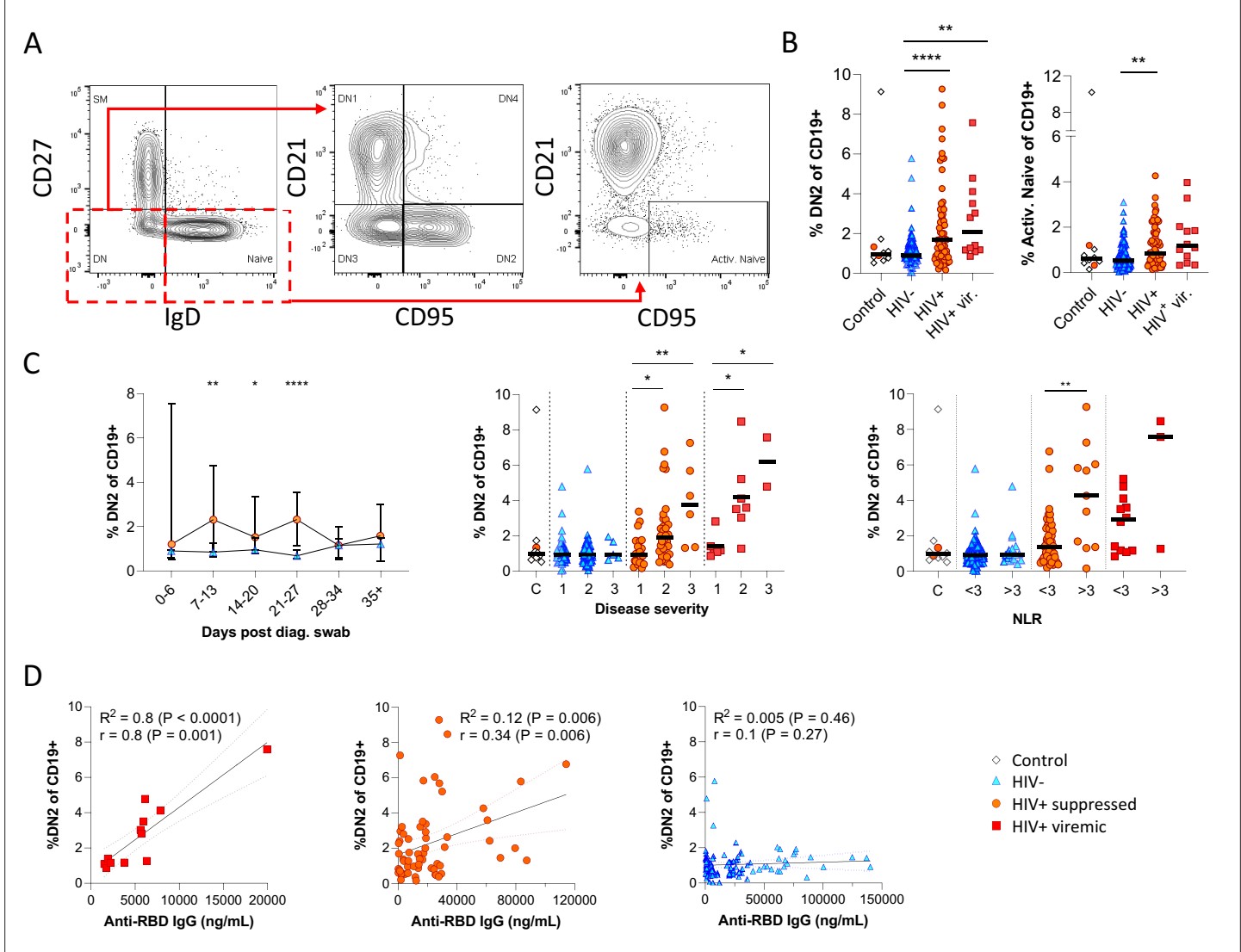

**Figure 4.** Pronounced extrafollicular B cell activation in people living with HIV (PLWH). (**A**) Naïve (CD27- IgD+) and double negative (DN; CD27- IgD-) B cell activation was measured as a CD21- CD95+ phenotype. The respective activated populations are thus DN2 and activated naïve. (**B**) Prevalence of the DN2 and activated naïve phenotypes with respect to HIV status. (**C**) The DN2 frequencies were tracked longitudinally and with respect to disease severity (ordinal scale 1–3) and neutrophil-lymphocyte ratio (NLR), respectively. Error bars indicate standard deviation (HIV- n = 37; HIV+ n = 33). (**D**) Spearman non-parametric correlation of the DN2 B cell response relative to the anti-receptor binding domain (RBD) antibody titre. Statistical analyses were performed using the Kruskal-Wallis H test for multiple comparisons and Mann-Whitney for disease severity or NLR comparisons within groups. p Values are denoted by *≤0.05; **<0.01; ***<0.001, and ****<0.0001.

The online version of this article includes the following figure supplement(s) for figure 4:

**Figure supplement 1.** Extended phenotyping of double negative (DN) B cells.

**Figure supplement 2.** Individual patient longitudinal DN2 and activated naïve B cell responses and detailed analysis of the activated naïve B cell response.

in HIV-ve participants is highlighted by the absence of DN populations even in subjects in OS3 and with elevated NLRs. These data are highly consistent with the data shown in *Figure 3* and suggest that the B cell response to SARS-CoV-2 infection in PLWH is associated with reduced GC maturation and increased EF activity.

Having observed elevated PD-L1 in non-COVID controls, we examined the expression of this marker longitudinally in conjunction with CD5 to examine regulatory B cell frequency (*Figure 5A*; *Figure 5—figure supplement 1*; *Catalán et al., 2021*; *Khan et al., 2015*; *Sun et al., 2019*). This

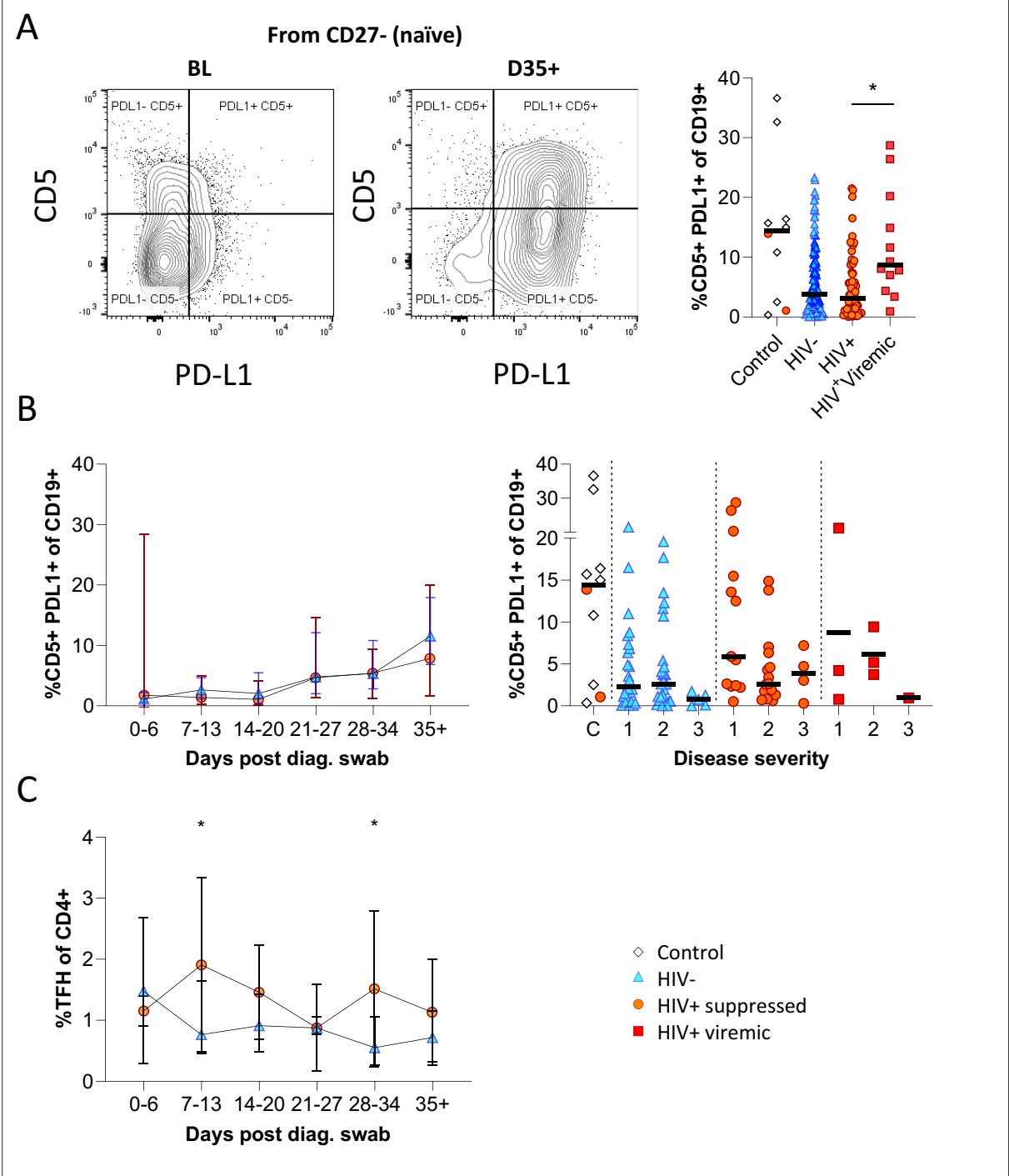

**Figure 5.** CD5+ PD-L1+ regulatory B cells contract during early response to infection. (**A**) Baseline (BL) and day 35 (D35+) example plots of a patient's CD5+ PD-L1+ regulatory B cell response. These cells were gated from the total naïve (CD27-) B cell population and their frequencies compared relative to HIV status. (**B**) This response was tracked longitudinally and relative to disease severity (ordinal scale 1–3) and controls denoted as 'C'. In (**C**) the corresponding CD4+ T follicular helper (Tfh) response was tracked longitudinally. Error bars indicate standard deviation (HIV- n = 37; HIV+ n = 33). Statistical analyses were performed using the Kruskal-Wallis H test for multiple comparisons and Mann-Whitney for disease severity. p Values are denoted by *≤0.05; **<0.01; ***<0.001, and ****<0.0001.

The online version of this article includes the following figure supplement(s) for figure 5:

**Figure supplement 1.** Individual patient longitudinal CD5+ PDL1+ B cell and T follicular helper (Tfh) responses.

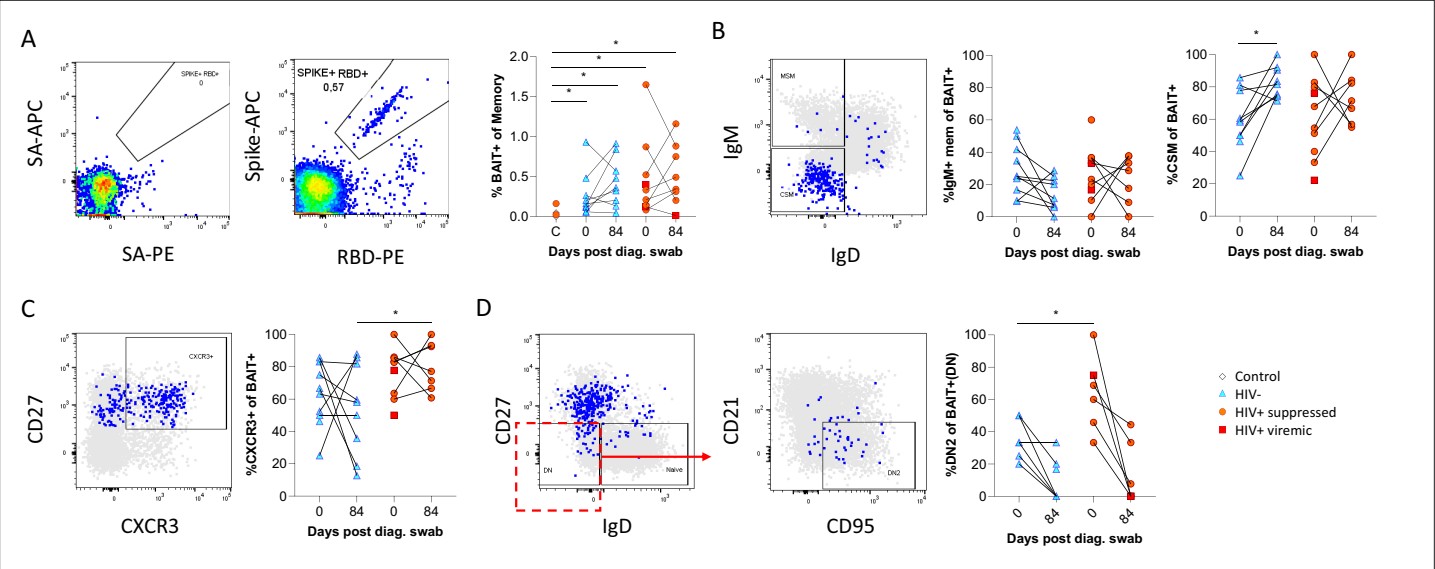

**Figure 6.** SARS-CoV-2 spike and receptor binding domain (RBD)-specific B cell responses highlight an upregulated extrafollicular response in people living with HIV (PLWH). The ancestral D614G viral spike (Spike-APC) and receptor binding domain (RBD-PE) proteins were used as baits to detect SARS-CoV-2-specific B cells with SA-APC and SA-PE used as controls (**A**). The bait-specific B cells were then overlaid onto an IgM vs. IgD plot (**B**). The extent of IgM-only memory and class switched memory (CSM) B cells were compared at both baseline (day 0) and day 84 post diagnosis. (**C**) Similarly, the level of CXCR3 expression was assessed. (**D**) The extent of double negative (DN) B cell activation (CD21- CD95+) was compared regarding HIV status at both time points. Statistical analyses were performed using Wilcoxon and Mann-Whitney tests. p Values are denoted by *≤0.05; **<0.01; ***<0.001, and ****<0.0001.

revealed a clear shift in PD-L1 expression on naïve B cells longitudinally, which was very low at baseline and increased to the range observed in controls by the final time point (*Figure 5B*). Similar frequencies and longitudinal trends were observed in both PLWH and HIV-ve participants, suggesting this is a consistent feature of the acute B cell response to SARS-CoV-2 infection. However, PD-L1 CD5+ B cells are significantly more frequent in HIV viremic individuals (*Figure 5A*). PD-L1 expression plays an integral part in the GC response and maintains the relatively unstable Tfh lineage (*Khan et al., 2015*). Interestingly, the frequency of total Tfh tended to be higher in PLWH, reaching significance at time points 2 and 5 (*Figure 5C*). Although the frequency of SARS-CoV-2-specific Tfh was not measured, these data again point to potential impairment of the GC response in PLWH, which is dependent on the crosstalk between B cells and Tfh governed, in part, by the interaction between PD-L1 expressed on B cells and PD-1 on Tfh (*Khan et al., 2015*; *Sun et al., 2019*).

Finally, as the data presented above was based on bulk B cell phenotyping, we examined a subset of informative markers on SARS-CoV-2-specific B cells using recombinant SARS-CoV-2 spike and RBD proteins conjugated to fluorescent streptavidin as baits (*Goel et al., 2021*;; *Figure 6A*; *Supplementary file 2*). B cells staining with both spike and RBD baits were quantified at baseline and 3 months, revealing robust SARS-CoV-2-specific memory B cell populations in all individuals, which did not significantly wane by 3 months and were not different in frequency between PLHW and HIV-ve participants (*Figure 6B*). Interestingly, in this subset of participants, the degree of CSM was not significantly different between groups and tended to increase over time in both. However, spike-specific memory B cells from PLWH tended to express higher levels of CXCR3 (*Figure 6C*), particularly at month 3, a marker associated with homing to inflamed tissue (*Onodera et al., 2012*; *Serre et al., 2012*; *Sutton et al., 2021*). Finally, a significantly higher proportion of spike-specific B cells from PLWH displayed a DN2 phenotype, confirming the increased EF activity towards SARS-CoV-2 in these individuals suggested by the bulk phenotyping.

## Discussion

Using longitudinal samples from the first wave of infection in South Africa, we found that HIV coinfection significantly impacted the B cell response to SARS-CoV-2. Overall, these data show that the B cell

response in PLWH is skewed towards an EF route and away from GC maturation. This is demonstrated by several observations, including elevated DN2 and activated naïve B cells in PLWH, consistent with EF maturation; mirrored by reduced class switching of memory B cells and reduced expression of markers CXCR5 and CD62L allowing B cells to home to the GC. This general reduction in GC homing capacity of B cells has been associated with PLWH previously (*Cagigi et al., 2008*). In addition, as an effective GC reaction requires tight regulation of the Tfh response, observed differences in PD-L1 expression on B cells and Tfh frequency in PLWH are likely to hamper B cell maturation via this pathway. Importantly, the skewing towards EF B cell maturation in PLWH correlated with anti-RBD antibody titre in these individuals and was confirmed on SARS-CoV-2 spike-specific B cells.

Multiple studies have characterized the B cell response to COVID-19 in HIV-ve individuals and observed a positive correlation between disease severity and an elevated ASC response (*Kaneko et al., 2020*; *Karim et al., 2021*; *Woodruff et al., 2020*). Furthermore, an EF B cell response has been associated with severe COVID-19 and predicts poor clinical outcomes, and severe COVID-19 cases have been characterized by poor GC formation in secondary lymphoid organs (*Chen et al., 2020*; *Kaneko et al., 2020*; *Woodruff et al., 2020*). However, the impact of HIV coinfection on the B cell response and these associations is unknown. Here, we find that the ASC response is associated with disease severity in both HIV-ve participants and PLWH, although the effect appears more robust in HIV-ve participants. More detailed phenotyping of the ASC supports this, as CD138+ PC and ASC with a tissue homing and activated phenotype (CXCR3+ CD69+ in panel 1 and CD86+ in panel 2) were more strongly associated with SARS-CoV-2 infection in HIV-ve participants. In contrast, the association between disease severity and EF activity was uniquely observed in PLWH. The absence of EF activity in HIV-ve participants is not at odds with published literature, as no individuals with severe COVID-19 were included in this study (*Kaneko et al., 2020*; *Woodruff et al., 2020*). Therefore, the association between HIV and EF B cells is not driven by disease severity.

Although not previously observed for COVID-19, the skewing of B cells towards an EF response in PLWH makes biological sense. Jenks et al. achieved successful ex vivo stimulation of DN2 B cells in response to pro-inflammatory cytokines IFNγ, TNFa, and IL-21; and TLR 7 and 9 stimulation, with CD40L having little effect, thus proposing a potentially extrafollicular maturation capacity of these cells, independent of T cell help (*Jenks et al., 2019*; *Jenks et al., 2020*). More recently Song et al. provided elegant in vivo evidence for peri-follicular interaction of DN2 B cells with Tfh, but not Th1 cells prior to GC formation and demonstrate that CD40L stimulation was integral to trigger their initial activation. Their model therefore suggests a peri-follicular, extra-GC maturation pathway requiring only an initial Tfh interaction (*Song et al., 2022*). HIV induces a pro-inflammatory state (*Connolly et al., 2005*; *Roff et al., 2014*), making B cells more prone to EF maturation; and HIV viremia is known to induce a DN2 response (*Amu et al., 2013*; *Ferreira et al., 2013*). This link may explain the association between DN2 frequency and inflammation in PLWH, as measured by the NLR ratio. On the other hand, since HIV depletes CD4 T cells, it also impairs GC activity, including BCR somatic hypermutation, class switching, and, ultimately, the ASC response (*Okoye and Picker, 2013*; *Pallikkuth et al., 2012*; *Perreau et al., 2013*), consistent with the trends observed. Likewise, HIV alters the B cell compartment by affecting the frequencies of naïve and memory B cells (*Moir and Fauci, 2014*), again agreeing with the differences in the frequency of naïve and memory B cells observed in this study.

The downstream consequence of skewed B cell maturation in PLWH is unclear from this study. However, the EF response relies primarily on the existing germline and memory BCR repertoire, whereas the GC response allows for honing of the BCR repertoire through somatic hypermutation and stringent affinity selection of BCR clones to generate high-affinity long-term ASC and memory responses (*Jenks et al., 2020*; *Kaneko et al., 2020*). Therefore, the loss of GC B cell maturation could result in a less effective B cell response to infection in PLWH and potentially a greater susceptibility to infection by variants. *Sette and Crotty, 2021*, demonstrated that the antibody response to COVID-19 parent strain derives from the germline BCR repertoire without the need for extensive hypermutation. This might explain why HIV status did not seem to affect the antibody response during the first wave of infections (*Snyman et al., 2021*). In contrast, the antibody response to the second wave of infections was affected by HIV status, with PLWH mounting less effective IgG responses to the Beta variant (*Hwa et al., 2022*). Therefore, the skewed EF B cell response could explain the less effective response against new variants. Indeed, multiple studies have revealed B cell maturation and expanded somatic hypermutation months after primary infection in COVID-19 patients without HIV

(*Gaebler et al., 2021*; *Wang et al., 2021*) and have highlighted the importance of antibody affinity maturation (*Chen et al., 2020*; *Muecksch et al., 2021*) and class switching (*Zohar et al., 2020*) to reduce disease severity and gain improved efficacy against new variants. This might also explain the lack of effective clearance of SARS-CoV-2 in HIV viremic individuals and might be a mechanism for intra-host evolution in patients with uncontrolled HIV (*Cele et al., 2022*). Further work is needed to understand how the skewed B cell response to natural infection impacts long-term memory and the ability to adapt to new viral variants. It will also be essential to understand the impact of vaccination on the B cell memory compartment.

## Additional information

### Group author details

**COMMIT-KZN Team**

**Moherndran Archary**: Department of Paediatrics and Child Health, University of KwaZulu-Natal, Durban, South Africa; **Kaylesh J Dullabh**: Department of Cardiothoracic Surgery, University of KwaZulu-Nata, Durban, South Africa; **Jennifer Giandhari**: KwaZulu-Natal Research Innovation and Sequencing Platform, Durban, South Africa; **Philip Goulder**: Africa Health Research Institute and Department of Paediatrics, Durban, South Africa; **Guy Harling**: Africa Health Research Institute and the Institute for Global Health, University College London, London, United Kingdom; **Rohen Harrichandparsad**: Department of Neurosurgery, University of KwaZulu-Natal, Durban, South Africa; **Kobus Herbst**: Africa Health Research Institute and the South African Population Research Infrastructure Network, Durban, South Africa; **Prakash Jeena**: Department of Paediatrics and Child Health, University of KwaZulu-Natal, Durban, South Africa; **Thandeka Khoza**: Africa Health Research Institute, Durban, South Africa; **Nigel Klein**: Africa Health Research Institute and the Institute of Child Health, University College London, London, United Kingdom; **Rajhmun Madansein**: Department of Cardiothoracic Surgery, University of KwaZulu-Nata, Durban, South Africa; **Mohlopheni Marakalala**: Africa Health Research Institute and Division of Infection and Immunity, University College London, London, United Kingdom; **Mosa Moshabela**: College of Health Sciences, University of KwaZulu-Natal, Durban, South Africa; **Kogie Naidoo**: Centre for the AIDS Programme of Research in South Africa, Durban, South Africa; **Zaza Ndhlovu**: Africa Health Research Institute and the Ragon Institute of MGH, MIT and Harvard, Durban, South Africa; **Kennedy Nyamande**: Department of Pulmonology and Critical Care, University of KwaZulu-Natal, Durban, South Africa; **Nesri Padayatchi**: Centre for the AIDS Programme of Research in South Africa, Durban, South Africa; **Vinod Patel**: Vinod Patel, Department of Neurology, University of KwaZulu-Natal, Durban, South Africa; **Theresa Smit**: Africa Health Research Institute, Durban, South Africa; **Adrie Steyn**: Africa Health Research Institute and Division of Infectious Diseases, University of Alabama, Birmingham, United States

### Competing interests

COMMIT-KZN Team: Alex Sigal: Reviewing editor, eLife. The other authors declare that no competing interests exist.

### Funding

| Funder | Grant reference number | Author |
| --- | --- | --- |
| Wellcome Trust | 201433/Z/16/Z | Alex Sigal |
| National Research Foundation | 64809 | Alex Sigal |
| Victor Daitz Foundation | | Alex Sigal |
| Max Planck Institute for Infection Biology | open access funding | Alex Sigal |

| Funder | Grant reference number | Author |
|--------|------------------------|--------|

The funders had no role in study design, data collection and interpretation, or the decision to submit the work for publication. For the purpose of Open Access, the authors have applied a CC BY public copyright license to any Author Accepted Manuscript version arising from this submission.

## Author contributions

Robert Krause, Conceptualization, Formal analysis, Investigation, Methodology, Writing - original draft, Writing – review and editing; Jumari Snyman, Hwa Shi-Hsia, Investigation, Methodology, Writing – review and editing; Daniel Muema, Writing – review and editing; Farina Karim, Khadija Khan, Matilda Mazibuko, Ntombifuthi Mthabela, Dirhona Ramjit, COMMIT-KZN Team, Project administration, Resources; Yashica Ganga, Abigail Ngoepe, Project administration, Methodology, Writing – review and editing; Yenzekile Zungu, Inbal Gazy, Methodology, Writing – review and editing; Mallory Bernstein, Project administration, Data curation, Writing – review and editing; Oliver Limbo, Joseph Jardine, Devin Sok, Ian A Wilson, Methodology, Resources, Validation; Willem Hanekom, Project administration, Funding acquisition, Writing – review and editing; Alex Sigal, Funding acquisition, Writing – review and editing; Henrik Kløverpris, Project administration, Writing – review and editing, Funding acquisition; Thumbi Ndung'u, Writing – review and editing, Funding acquisition; Alasdair Leslie, Conceptualization, Supervision, Funding acquisition, Writing – review and editing

## Author ORCIDs

Robert Krause (ID) http://orcid.org/0000-0003-1558-0397
Farina Karim (ID) http://orcid.org/0000-0001-9698-016X
Khadija Khan (ID) http://orcid.org/0000-0001-7565-7400
Alex Sigal (ID) http://orcid.org/0000-0001-8571-2004
Thumbi Ndung'u (ID) http://orcid.org/0000-0003-2962-3992

## Ethics

Human subjects: The study protocol was approved by the University of KwaZulu-Natal Biomedical Research Ethics Committee (approval BREC/00001275/2020). Written informed consent was obtained for all enrolled participants.

## Decision letter and Author response

Decision letter https://doi.org/10.7554/eLife.79924.sa1
Author response https://doi.org/10.7554/eLife.79924.sa2

# Additional files

## Supplementary files
- Supplementary file 1. Flow cytometry B cell phenotyping antibody panels.
- Supplementary file 2. Flow cytometry B cell BAIT antibody panel.
- MDAR checklist
- Source data 1. Raw population frequency data for *Figures 1–6*.
- Reporting standard 1. STROBE observational study checklist.

## Data availability

All data generated or analyzed during this study are included in the manuscript and Source data 1.

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
