## [Editor Report]

The authors investigate how HIV-1 infection affects the immune response in the context of SARS-CoV-2 infection by characterising the circulating B cell response. They conclude that people with HIV-1 infection, who become infected by SARS-CoV-2, produce B cell responses via an extra-follicular pathway to a greater degree than people who do not have HIV-1 infection. These findings imply that in HIV-1 infected individuals, long-term B cell and antibody responses against SARS-CoV-2 might not be as robust and durable compared to those in people without HIV-1 infection. The manuscript will be of interest to infectious disease specialists, virologists, and immunologists.

---

## [Decision Letter]

**Decision letter after peer review:**

Thank you for submitting your article "HIV skews the SARS-CoV-2 B cell response toward an extrafollicular maturation pathway" for consideration by *eLife*. Your article has been reviewed by 2 peer reviewers, and the evaluation has been overseen by a Reviewing Editor and Bavesh Kana as the Senior Editor. The following individuals involved in review of your submission have agreed to reveal their identity: Martyn Andrew French (Reviewer #2); Zheng Zhang (Reviewer #3).

The reviewers have discussed their reviews with one another, and the Reviewing Editor has drafted this to help you prepare a revised submission. These points have been summarised below, please refer to the reviewer comments further below for more detail.

Essential revisions:

1. There are concerns about nomenclature of cell populations defined by tSNE plots (figure 2A). For example, the population defined as "CSM/marginal zone" does not express IgD or IgM, as would be expected for class-switched memory B cells but not marginal zone B cells. In addition, while tissue homing and GC homing CSM B cells express expected amounts of CXCR4 and CXCR5, both express high amounts of CXCR3, which would be unexpected for GC homing cells. Finally, in line 144, the authors should clarify what is meant by "class switched, IgMhi B cells (highlighted in blue)". The population highlighted in blue in figure 2A, referred to as "IgM++ GC homing B cells", has the immunophenotype IgDlow, IgMhigh, CD27-. Aren't these cells at one end of a naïve B cell spectrum ranging from IgD+/IgM- to IgD+/IgM+ to IgDlow /IgMhigh? There are also other populations that have unconventional names and/or appear to be intermediary populations.

2. The authors have defined DN2 B cells based on expression of the activation marker CD95 (Fas) (see Figure 4) but the original definition of DN2 B cells in patients with SLE was based on expression of CD11c and lack of expression of CXCR5 (see – Jenks SA et al. Immunity. 2020; 52:203). These cells also express T-bet and therefore, have many characteristics in common with CD11c+/T-bet+ memory B cells (also known as age-associated B cells or atypical memory B cells). It would be informative if data on CXCR5- DN B cells were in analysed in addition to, or instead of, CD95+ DN B cells.

3. It might also be informative to discuss the extra-follicular (EF) response pathway in more detail. Recently published data from studies undertaken in mice indicate that CD11c+/T-bet+ MBCs interact with T follicular helper cells in lymphoid follicles but not in germinal centres (Song W et al. Immunity 2022; 55:290-307.e5), so it could be argued that the differentiation pathway is extra-GC rather than extra-follicular, at least in some situations. Also, in people with HIV-1 infection, HIV-1 gp140-specific B cells expressing T-bet are produced outside of GCs (Austin JW et al. Sci Transl Med. 2019; 11:eaax0904). Is the EF response pathway different to the extra-GC differentiation pathway? Where does it occur?

4. Similarly, in lines 288-290, the authors should re-consider the statement that "Both DN2 and activated naïve B cells mature via an EF pathway, independent of T cell help and in response to pro-inflammatory cytokines IFNγ, TNFa, and IL-21; and TLR 7 and 9 stimulation". There are data indicating that differentiation of DN2 B cells is T-cell-dependent (Keller B et al. Sci. Immunol. 2021; 6:eabh0891).

5. In lines 254-60 and figure 6, the investigators should consider the possibility that the CXCR3+ and DN2 SARS-CoV-2-specific MBCs that are increased in people with HIV infection are the same population of cells. CD11c+/T-bet+ MBCs (ie. DN2 B cells, age-associated B cells or atypical memory B cells) usually express high levels of CXCR3.

6. The dynamic changes of cell proportion at different timepoints such as ASC, DN2 were shown in this paper. Can the authors display the dynamic changes of SARS-CoV-2 viral load matched to the cell ratio to demonstrate the association between alteration of cell ratio and virus infection?

7. Since the control group was comprised of healthy participants and HIV infected individuals, the control group should be sub-classified into HIV+ and HIV- groups, and all the results should be shown separately. In this way, the cell ratio in PLWH at baseline (not infected with SARS-CoV-2), and the contribution of SARS-CoV-2 or HIV infection on cell ratio alteration could be clearly shown.

8. It has been reported that DN2 cells have dampened BCR signalling and antibody secreting capacity in HIV infected individuals but might play the converse role during acute infection. Only the increased cell ratio of DN2 was revealed in PLWH post SARS-CoV-2 infection, the antibody secreting ability of DN2 in vitro was not demonstrated. This limits conclusions that can be made. Can the authors comment?

Other comments that must be addressed:

1. In lines 266-7,do the authors mean that reduced expression of CXCR5 and CD62L would prevent B cells homing to the GC?

2. IgM switched memory B cells (lines 201-207) are referred to as IgM-only memory B cells by some investigators (for example, see – Bautista D et al. Front Immunol. 2020; 11:736). It would help the reader if this were indicated.

3. Please note the timepoints when cell ratio detected in Figure 1B and Figure 2. It is important to understand which stages early or late was mostly influenced by HIV infection induced DN2 accumulation.

*Reviewer #1 (Recommendations for the authors):*

The authors should consider revising terminology used to define B cell subpopulations defined in the tSNE plots. Also, there needs to be further clarification of how the authors' definitions of DN2 MBCs fits in with other definitions of DN2 MBCs and identical or similar cells, ie. CD11c+/T-bet+ MBCs, age-associated B cells, atypical MBCs etc.

*Reviewer #2 (Recommendations for the authors):*

There are still some questions need attention.

1. The dynamic changes of cell proportion at different timepoints such as ASC, DN2 were shown in this paper. I suggest you display the dynamic changes of SARS-CoV-2 viral load matched to the cell ratio to demonstrate the association between alteration of cell ratio and virus infection.

2. Since the control group was comprised of healthy participants and HIV infected individuals, control group should be sub-classified into HIV+ and HIV- groups, and all the results should be shown separately. In this way, the cell ratio of PLWH at baseline (not infected with SARS-CoV-2), and the contribution of SARS-CoV-2 or HIV infection on cell ratio alteration should be clearly shown.

3. It has been reported that DN2 cells had dampened BCR signaling and antibody secreting capacity in HIV infected individuals, but might play the conversed role during acute infection. Only the increased cell ratio of DN2 were revealed in PLWH post SARS-CoV-2 infection, the antibody secreting ability of DN2 in vitro lacked. The results in the manuscript cannot support the conclusion that PLWH displayed the reduced ASC responses.

4. The RBD specific antibody concentration in serum from HIV-ve and PLWH also needs to detected. Although the DN2 cells accumulated in PLWH, the amount of antigen specific memory B cells were comparable between HIV-ve and PLWH during both infected and convalescent stages, the contribution of DN2 accretion in RBD specific antibody production needs further specified.

5. Please note the timepoints when cell ratio detected in Figure 1B and Figure 2. It is important to understand which stages early or late was mostly influenced by HIV infection induced DN2 accumulation.

---

## [Author Response]

Essential revisions:1. There are concerns about nomenclature of cell populations defined by tSNE plots (figure 2A). For example, the population defined as "CSM/marginal zone" does not express IgD or IgM, as would be expected for class-switched memory B cells but not marginal zone B cells. In addition, while tissue homing and GC homing CSM B cells express expected amounts of CXCR4 and CXCR5, both express high amounts of CXCR3, which would be unexpected for GC homing cells. Finally, in line 144, the authors should clarify what is meant by "class switched, IgMhi B cells (highlighted in blue)". The population highlighted in blue in figure 2A, referred to as "IgM++ GC homing B cells", has the immunophenotype IgDlow, IgMhigh, CD27-. Aren't these cells at one end of a naïve B cell spectrum ranging from IgD+/IgM- to IgD+/IgM+ to IgDlow /IgMhigh? There are also other populations that have unconventional names and/or appear to be intermediary populations.

Regarding the nomenclature used in Figure 2A, we thank the reviewer for these points and have made the following revisions to the figure (originally on pg 18) and in text as listed.

CSM/marginal zone phenotype was corrected to CSM, since, as the reviewer correctly pointed out, there was not a significantly increased expression of IgM detected on these cells.

Elevated expression of CXCR3 in class switched memory GC B cells was suggestive of transition/maturation to plasma blasts/plasma cells (Muehlinghaus *et al.*, 2005, *Blood*). This was referred to in text in line 214, pg 5.

Class switched IgM high nomenclature was changed to IgM high naïve. This was referred to in text in line 207, pg 5.

The Transitional/naïve phenotype expressing high CCR6 and IgD was renamed to pre-GC naïve (Schwickert *et al.*, 2011, *JEM*) and referred to in text in line 221 pg 6.

2. The authors have defined DN2 B cells based on expression of the activation marker CD95 (Fas) (see Figure 4) but the original definition of DN2 B cells in patients with SLE was based on expression of CD11c and lack of expression of CXCR5 (see – Jenks SA et al., Immunity. 2020; 52:203). These cells also express T-bet and therefore, have many characteristics in common with CD11c+/T-bet+ memory B cells (also known as age-associated B cells or atypical memory B cells). It would be informative if data on CXCR5- DN B cells were in analysed in addition to, or instead of, CD95+ DN B cells.

We agree with the reviewer regarding the standard definition of the DN2 phenotype as described by Jenks et al., Immunity. 2020. During the optimization process, we confirmed that CD95 also served as an excellent marker of DN2 but did not include this in the original submission. We have now included this data in extended data to Figure 4 (Extended Data Figure 4.1) on pg 29, which demonstrates that the CD21- cells (on which CD95 is sub-gated) lack expression of CXCR5, supporting that this is a similar population studied/referred to by Jenks *et al.*, (2020). We also measured the overlap between CD11c expression and CD95 for a smaller subset of participants, which resulted in a positive correlation (R^2^ = 0.66, P < 0.0001). This was referred to in text in line 285, pg 7.

The following legend was included, line 508, pg 12: Figure 4—figure supplement 1. Extended phenotyping of DN B cells. Gating DN B cells for CD21 versus CXCR5 revealed two dominant populations being either double positive or negative for both markers (A). CD11c and CD95 surface expression on DN B cells correlated positively (B). The frequency of DN2 and active naïve B cells were related to SARS-CoV-2 viral load (C).

We have renamed the original Extended Data Figure 4, which displays the activated naïve phenotype to Figure 4—figure supplement 2 (originally on pg 30) and have amended the figure legend accordingly, line 512, pg 12 and the in text references in line 289, 292, 300 and 302 pg 7.

3. It might also be informative to discuss the extra-follicular (EF) response pathway in more detail. Recently published data from studies undertaken in mice indicate that CD11c+/T-bet+ MBCs interact with T follicular helper cells in lymphoid follicles but not in germinal centres (Song W et al., Immunity 2022; 55:290-307.e5), so it could be argued that the differentiation pathway is extra-GC rather than extra-follicular, at least in some situations. Also, in people with HIV-1 infection, HIV-1 gp140-specific B cells expressing T-bet are produced outside of GCs (Austin JW et al., Sci Transl Med. 2019; 11:eaax0904). Is the EF response pathway different to the extra-GC differentiation pathway? Where does it occur?

We thank the reviewer for this comment and have addressed it with the point below.

4. Similarly, in lines 288-290, the authors should re-consider the statement that "Both DN2 and activated naïve B cells mature via an EF pathway, independent of T cell help and in response to pro-inflammatory cytokines IFNγ, TNFa, and IL-21; and TLR 7 and 9 stimulation". There are data indicating that differentiation of DN2 B cells is T-cell-dependent (Keller B et al., Sci. Immunol. 2021; 6:eabh0891).

We thank the reviewer for these insightful references and have made amendments to the text to include further discussion of the extrafollicular pathway (point 3) and the maturation of DN2 B cells (point 4) and have added these to the text, line 363 pg 9.

Jenks *et al.,* achieved successful ex vivo stimulation of DN2 B cells in response to pro-inflammatory cytokines IFNγ, TNFa, and IL-21; and TLR 7 and 9 stimulation, with CD40L having little effect, thus proposing a potentially extrafollicular maturation capacity of these cells, independent of T cell help (Jenks et al., 2019; Jenks et al., 2020). More recently Song *et al.,* provided elegant in vivo evidence for peri-follicular interaction of DN2 B cells with T follicular helper, but not Th1 cells prior to germinal center (GC) formation and demonstrate that CD40L stimulation was integral to trigger their initial activation. Their model therefore suggests a peri-follicular, extra-GC maturation pathway requiring only an initial Tfh interaction (Song *et al.*, 2022).

5. In lines 254-60 and figure 6, the investigators should consider the possibility that the CXCR3+ and DN2 SARS-CoV-2-specific MBCs that are increased in people with HIV infection are the same population of cells. CD11c+/T-bet+ MBCs (ie. DN2 B cells, age-associated B cells or atypical memory B cells) usually express high levels of CXCR3.

This is a legitimate concern, given the partially overlapping phenotypes. However, in the case of Figure 6, the population in question is CD27 expressing memory B cells, whilst the DN B cell phenotype are strictly CD27 negative (CD27- IgD-). We are therefore confident that there is no overlap in these populations and, therefore, that the CXCR3 expression measured here does not derive from the DN2 B cells.

6. The dynamic changes of cell proportion at different timepoints such as ASC, DN2 were shown in this paper. Can the authors display the dynamic changes of SARS-CoV-2 viral load matched to the cell ratio to demonstrate the association between alteration of cell ratio and virus infection?

We thank the reviewer for this comment. The association between the DN2 phenotype and SARS-CoV-2 viraemia has now been included in Figure 4—figure supplement 1C. The text was amended from line 294 pg 7.

Interestingly though, a direct association with SARS-CoV-2 viral load was evident only in the PLWH with suppressed HIV, whereas for HIV viremic individuals, the DN2 phenotype associated better with disease severity and inflammation (Figure 4—figure supplement 1C), which has been associated with chronic HIV infection (Austin et al., 2019).

7. Since the control group was comprised of healthy participants and HIV infected individuals, the control group should be sub-classified into HIV+ and HIV- groups, and all the results should be shown separately. In this way, the cell ratio in PLWH at baseline (not infected with SARS-CoV-2), and the contribution of SARS-CoV-2 or HIV infection on cell ratio alteration could be clearly shown.

We agree with the reviewer that this would have been an interesting comparison. Unfortunately, we only had two HIV positive participants within the total control group of n = 10. To highlight these two participants however, we have opted to use the same symbol and colour as that for the HIV+ suppressed in the other figures and have changed these in all figures concerned (Figures 1, 3, 4, 5, Figure 4—figure supplement 1 and Figure 4—figure supplement 2).

8. It has been reported that DN2 cells have dampened BCR signalling and antibody secreting capacity in HIV infected individuals but might play the converse role during acute infection. Only the increased cell ratio of DN2 was revealed in PLWH post SARS-CoV-2 infection, the antibody secreting ability of DN2 in vitro was not demonstrated. This limits conclusions that can be made. Can the authors comment?

This is a valid point raised by the reviewer. Although we have not done any in vitro mechanistic testing ourselves, there was a positive correlation (R = 0.8, P = 0.001) between the DN2 phenotype in the HIV viraemic patients and anti-SARS-CoV-2 specific RBD binding antibodies (Figure 4D), suggestive of a causal association.

Other comments that must be addressed:1. In lines 266-7,do the authors mean that reduced expression of CXCR5 and CD62L would prevent B cells homing to the GC?

We add the following statement to the discussion in line 340 pg 8: This general reduction in GC homing capacity of B cells has been associated with PLWH previously (Cagigi et al., 2008).

There seems to be a reduction in the capacity of B cells to home to germinal centres in PLWH compared to HIV negative participants. This is based on the fact that there is a reduction in the overall expression of CXCR5 and CD62L. Longitudinally the levels are reduced but not significantly different between the two groups, with the only significant difference being between HIV negative patients with viraemic HIV overall. This is in keeping with what has been reported previously (Cagigi *et al.*, 2008) for PLWH versus HIV negative controls.

2. IgM switched memory B cells (lines 201-207) are referred to as IgM-only memory B cells by some investigators (for example, see – Bautista D et al., Front Immunol. 2020; 11:736). It would help the reader if this were indicated.

We thank the reviewer for this comment and have changed the reference to this phenotype to IgM-only memory and included the Bautista reference as suggested in line 203 pg 5. The axis label in Figure 3C was changed from MSM to IgM+ mem of CD19^+^ and the figure legend changed accordingly, line 455 pg 11. Similarly Figure 6 was amended together with its figure legend (line 481 pg 11).

3. Please note the timepoints when cell ratio detected in Figure 1B and Figure 2. It is important to understand which stages early or late was mostly influenced by HIV infection induced DN2 accumulation.

We thank the reviewer for this comment. These were clarified in text and relevant figure legends.

In Figure 1B the total frequencies at all time points were combined and compared between patient groups to gain an overview of the general frequencies. For those with longitudinal trends, or trends based on clinical parameters, these were then stratified. This was outlined in the figure legend line 430 pg 10.

The time points used to compile Figure 2 were all at baseline, as this figure compares differences in disease severity, SARS-CoV-2 viral load and neutrophil lymphocyte ratio at the earliest time point following positive diagnosis. Accordingly this was now stated in the Results section in line 193 pg 5 and in the Figure legend, line 439 pg10.